# Dynamics of Milk Parameters of Quarter Samples before and after the Dry Period on Czech Farms

**DOI:** 10.3390/ani13040712

**Published:** 2023-02-17

**Authors:** Lucie Kejdova Rysova, Jaromir Duchacek, Veronika Legarova, Matus Gasparik, Anna Sebova, Sona Hermanova, Radim Codl, Jan Pytlik, Ludek Stadnik, Hana Nejeschlebova

**Affiliations:** 1Department of Food Science, Faculty of Agrobiology, Food and Natural Resources, Czech University of Life Sciences Prague, Kamycka 129, 16500 Prague-Suchdol, Czech Republic; 2Department of Animal Science, Faculty of Agrobiology, Food and Natural Resources, Czech University of Life Sciences Prague, Kamycka 129, 16500 Prague-Suchdol, Czech Republic; 3Dairy Research Institute Ltd., Ke Dvoru 12a, 16000 Prague 6-Vokovice, Czech Republic

**Keywords:** dairy cow, dry period, milk, selective dry cow therapy, somatic cell count

## Abstract

**Simple Summary:**

One of the most significant global health threats to humanity is the loss of the therapeutic effects of antibiotics, which several factors, such as the overuse of antibiotics in livestock farming, can cause. The European Union has not let this threat go unnoticed and has decided to regulate the use of antibiotics in farm animals with new regulations. For this reason, we decided to monitor selected milk parameters such as fat, protein, casein, lactose, solids-not-fat content, total solids content, freezing point, titratable acidity, somatic cell count, and influences such as parity, farm, day of calving, and time of evaluation to identify how prepared Czech dairy farms are for the selective dry cow therapy. Our results indicate that exploiting somatic cell count is one of the possibilities for implementing this therapy. However, well-known mastitis prevention and control strategies must be in place on farms, and the content of some of the leading milk components could also be considered.

**Abstract:**

This study aimed to monitor milk parameters on three different dairy farms in the Czech Republic to describe their readiness for implementing selective dry cow therapy. Fat, protein, casein, lactose, solids-not-fat content, total solids content, freezing point, titratable acidity, and somatic cell count of quarter milk samples collected from tested Holstein cows were evaluated. Associations between the tested parameters, as well as the effects of parity, farm, day of calving, and time of evaluation at dry-off and after calving, were assessed. Values of the leading milk components dynamically changed between dry-off and after calving, but only protein content was significantly affected. The most important parameter of our research, the somatic cell count of quarter milk samples, was also not affected by the time of evaluation. Even though a slight increase in the mean of somatic cell count is expected before the dry period and after calving, at dry-off, we observed 30%, 42%, and 24% of quarters with somatic cell counts above 200,000 cells per mL, while after calving, we observed 27%, 16%, and 18% of quarters with somatic cell counts above 200,000 cells per mL on Farm 1, Farm 2, and Farm 3, respectively. High somatic cell counts (>200,000 cells per mL) indicate bacterial infection, as confirmed by the significant negative correlation between this parameter and lactose content. In addition, a deficient milk fat-to-protein ratio was observed on two farms, which may indicate metabolic disorders, as well as the occurrence of intramammary infections. Despite the above, we concluded that according to the thresholds of somatic cell counts for selective dry cow therapy taken from foreign studies, a large part of the udder quarters could be dried off without the administration of antibiotics. However, it is necessary to set up more effective mechanisms for mastitis prevention.

## 1. Introduction

Somatic cell count (SCC) is a worldwide parameter commonly used to evaluate milk quality and udder health. The determination of this parameter is essential not only from a technological point of view but also from a food safety point of view, where a high SCC may involve significant risks to consumers’ health [1,2]. The limit for SCC was first imposed in the United States in 1967. Subsequently, the directive on hygiene rules for raw milk was adopted in 1992 in the EU [3]. Currently, the limit of SCC in the European Union, including the Czech Republic, is regulated by Regulation 2004/853, which states that raw bovine milk must have an SCC lower than or equal to 400,000 cells per 1 mL of milk. The assessed value of the SCC shall be determined as a rolling geometric average over three months, with at least one sample per month [4].

Another Regulation of the European Union (Regulation 2019/6) that is likely to have a significant impact on the management of dairy farms entered into force in January 2022 [5]. Regulation 2019/6 will prevent the routine prophylactic and metaphylactic use of antibiotics in the coming years. Dairy farms will particularly feel this change in the transition to the dry period, where blanket dry cow therapy use will only be possible with prior veterinary justification. The urgency of improving treatment strategies in livestock is evidenced by data reporting an increase in antibiotic consumption and the presence of multi-resistant bacteria in milk samples across Europe [6]. The need for antibiotics on dairy farms will also have to be minimized in the Czech Republic. Indeed, the downward trend in the use of intramammary antibiotics at dry-off stopped, followed by a severe increase of 20% between 2016 and 2019. The data also show that of the 361,430 dairy cows, more than 80% were dried off by intramammary antibiotics in 2019 [7].

The above shows that selective dry cow treatment is scarcely applied in the Czech Republic. This is probably due to the pitfalls that accompany the introduction of selective dry-off on dairy farms. One of them is that transitioning from the lactating state to the non-lactating state is a high-risk period for intramammary infection [8]. Consequently, the primary challenge implementation of selective dry cow therapy is to set selection criteria separating cows or udder quarters according to the presence or suspicion of intramammary infection into two groups: treatment with or without antibiotics [9,10]. One of the methods to determine the selection criterion is the SCC mentioned above. Thus, some European countries have established thresholds or flowcharts based on SCC to identify cows suitable for selective dry cow treatment [11,12]. However, not only SCC can be a reliable indicator of inflammatory processes in the udder. For example, it is known that lactose levels can be used for the early identification of metabolic disorders and mastitis [13]. The biochemical processes resulting from inflammatory infection also affect other components of the milk solids [14]. Therefore, the possibility of using mid-infrared spectroscopy not only for the routine determination of the main components of milk but also for the incidence of mastitis is currently being tested [15]. As milk quality in the Czech Republic is increasing steadily in terms of hygiene and health indicators (composition, count of microorganisms, SCC, and residues of inhibiting substances), we hypothesized that selective dry cow therapy could be applied to part of the quarters. To investigate this, the aims of this study were to measure SCC and other milk parameters before and after the dry period at the level of individual quarters, assess the correlation between these values, and explore the influence of parity, farms, time of evaluation, and day of calving to understand the possible variations in the parameters better.

## 2. Materials and Methods

In total, 386 quarter milk samples were collected and evaluated from 3 dairy farms (Farm 1 = 84 samples; Farm 2 = 114 samples; Farm 3 = 188 samples) in the Czech Republic from December 2021 to June 2022. The samples were collected in two distinct periods: at the dry-off (n = 211) and after calving (n = 175), while one cow had one non-functional teat and was only milked from 3 teats. The distribution of cows based on lactation number corresponded to the profile of the herds, when 21 cows entered the test at the first lactation, 14 cows at the second lactation, and 18 cows at the third and higher lactation. Some of the cows sampled at the dry-off were culled during the dry period, and a few were culled due to post-partum health problems (n = 9). This cohort was selected according to the typical breeding profile in the Czech Republic, where the Holstein-Friesians breed predominates [16]. The farms were located in two Czech regions, and their herds ranged from 100 to 400 cows. All farms used parlors to obtain milk. Dairy cows were included in the study, with no consideration of lactation order, mastitis incidence, or milk yield. Part of the quarter samples was taken on the day of drying-off, and part was taken from the same cows between 6 and 21 d in milk. Blanket dry cow therapy was the standard way to dry off cows on all farms. The length of the dry period was around 60 d for all quarters of all cows. The geometric mean of the bulk milk somatic cell count on each farm during the sampling period was: Farm 1—155,000 cells per mL; Farm 2—148,000 cells per mL; and Farm 3—219,000 cells per mL. Approximately 100 mL of quarter milk samples were collected into 50 mL Falcon tubes (VWR International, Radnor, PA, USA). Before collection, each teat was cleaned with a paper cloth soaked in 70% ethanol. Then, 3 to 4 squirts were performed to remove the milk with the highest bacterial contamination from the teat. The sampling procedure reflected the sampling principles specified in the methodology of performance control for the Czech dairy farms. All samples recorded the cow`s identification number, lactation order, and teat position. Milk samples were transported from farms under refrigerated conditions (<6 °C) to the Milk Laboratory at the Czech University of Life Sciences, Prague.

The raw milk samples were homogenized by an IKA MS 3 instrument and warmed to 40 °C in a water bath. Then, the fat, protein, casein, lactose, total solids (TS), solids-not-fat (SNF), and freezing point (FP) of the milk were determined using a MilkoScan FT 120 (FossElectric, Hillerød, Denmark). The analyzer was regularly calibrated for the dairy components of raw cow milk. The titratable acidity of raw milk samples was determined with the 0.25 M NaOH standard solution titration method, where 10 mL of samples with phenolphthalein as the indicator were mixed and titrated to the equivalence point. The results were calculated as Soxhlet–Henkel degrees (°SH). The SCC of milk samples was measured by a Lactoscan milk SCC counter by the fluorescent microscope technique of cell counting (Milkotronic Ltd., Nova Zagora, Bulgaria). Approximately 100 µL of tempered and homogenized milk sample was added to the microtube with Sofia Green dye. This mixture was then homogenized, and 8 µL was put into one of the four chambers of the cartridge (Lachtochip 4R 50 µm 4 × 16, Milkotronic Ltd., Nova Zagora, Bulgaria). This procedure was repeated four times with each of the quarter samples. The results were expressed in cells/mL.

The statistical analyses were performed with the SAS software ver. 9.4 (SAS/STAT; SAS Institute, Inc. Cary, NC, USA). The first descriptive statistics were calculated by the UNIVARIATE procedure, including arithmetical mean, minimum, maximum, standard error, and coefficient of variation. Then, to measure the tightness of the relationship between the analyzed variables, Pearson correlation coefficients were determined using the CORR procedure. The correlation coefficients were evaluated as follows: 0.30 to 0.59 (−0.30 to −0.59), fair; 0.60 to 0.79 (−0.60 to −0.79), moderately strong; and at least 0.80 (−0.80), very strong. Values of the correlation coefficient that were less than +0.30 (−0.30) were indicated as a poor correlation regardless of the significance level. Therefore, they were not commented on in the results [17]. Multifactor analysis of variance was explored using the GLM procedure, where milk parameters were coded as dependent variables. The REG procedure, the STEPWISE method, and the Akaike information criterion parameter were used to select appropriate effects for the model equation. The model equation included the fixed effects of parity, time of evaluation, farm, and regression on the day of calving. The fourth effect of teat position was also tested, which was subsequently discarded due to statistical non-significance for all experimental parameters (*p* > 0.05). Differences between means were evaluated using the Tukey–Kramer test.

Multifactorial analysis of variance was applied according to the following model equation:Y_ijkl_ = µ + PAR_i_ + TIME_j_ + FARM_k_ + b × (day of calving) + e_ijkl_;(1)
where Y_ijkl_ represents the measured value of dependent variables (SCC (cells/mL); titratable acidity (°SH); fat (%); protein (%); casein (%); lactose (%); TS content (%); SNF content (%); FP (°C)); µ represents the mean value of the dependent variable; PAR_i_ represents the fixed effect of parity (i = 1, n = 84; i = 2, n = 115; i = 3, n = 187); TIME_j_ represents the fixed effect of time of evaluation (j = before calving, n = 211; j = after calving, n = 175); FARM_k_ represents the fixed effect of a farm (Farm 1, n = 84; Farm 2, n = 114; Farm 3, n = 188); b × (day of calving) represents the linear regression on the day of calving; and e_ijkl_ represents the random residual error. The significance levels were set at *p* < 0.05 and *p* < 0.01.

## 3. Results

Descriptive statistics for the investigated milk parameters are reported in Table 1. The resulting values are based on a dataset including all three farms participating in the experiment. An arithmetical mean, minimum, maximum, standard error and coefficient of variation are given for milk samples at dry-off and after calving. The means of fat, protein, and casein at dry-off were considerably higher than after calving. This downward trend was logically followed by means of SNF and TS. The difference in milk fat and protein after calving was accompanied by increased SCC from 170,659 in 1 mL to 254,637, despite blanket dry cow therapy with intramammary antimicrobials being carried out on all farms.

In contrast, lactose content increased after calving. The analyzed physical properties of the milk (FP and titratable acidity) changed minimally before and after the dry period. In terms of variability, as shown in Table 1, the highest coefficients of variation before and after calving were observed for fat and SCC. The coefficient of variation of the SCC was greater than 100% in both cases. After calving, this hygiene indicator even reached a value of 362.54%, with extreme values of its minimum (3000) and maximum (7,600,000). Other monitored parameters showed a lower variability before the dry period and after calving (CV < 20%). The fat-to-protein ratio after calving on the farms included in the experiment was as follows: Farm 1—0.5; Farm 2—1.2; Farm 3—0.6.

Table 2 lists the significant and non-significant correlation between the parity, main components of milk, FP, titratable acidity, and SCC before the dry period. It is clear from the table that parity was significantly correlated with some milk parameters, but these were only poor correlation coefficients. A fair negative correlation was confirmed between fat and lactose content. Total protein at dry-off correlated very strongly only with casein content from the major components. SNF content was strongly positively correlated with protein, casein, and lactose, while TS content strongly correlated with fat and weakly correlated with protein and casein. The physical properties also showed significant positive correlations. FP was correlated with protein, casein, lactose, SNF, and TS content. The strongest correlation was observed between FP and SNF. Additionally, the titratable acidity is weakly correlated with casein, lactose, and SNF. The correlations between SCC and other observed parameters (fat, SNF content, FP, titratable acidity) showed only poor correlation coefficients (r = from −0.3 to 0.3), except for lactose (r = −0.355; *p* < 0.01).

We found that some results remained unchanged when assessing the correlation between parameters after calving, as seen in Table 3. No strong correlations were found between parity and milk components and physical properties. The same results were obtained for SCC. A fair and strong correlation has been demonstrated between the contents of the leading milk components. The strongest Pearson’s correlation was observed between total protein and casein. Lactose content negatively correlated with fat content, as it was at the dry-off. In addition, a negative correlation was found between lactose and total protein. As expected, SNF content was positively correlated with leading milk components such as protein and lactose and was negatively correlated with fat. Total protein and casein were not strongly correlated to TS content after calving, whereas the correlation between fat content and TS content remained. A negative correlation between TS content and lactose was also observed after calving. The physical properties were correlated with some milk components and with each other.

The GLM procedure confirmed the significant (*p* < 0.05) influence of the effects on almost all analyzed milk parameters, except for SCC, as seen in Table 4. For SCC, the model explained only 2.7% of the variability, and a similar value of r^2^ was recorded for titratable acidity. The model explained between 14.3% and 48.1% of the variability for other milk parameters. The fixed effects of parity, farm, and regression on the day of calving were statistically significant for several different milk parameters, unlike the effect of time of evaluation, which was only significant for protein and FP (*p* < 0.05).

A more detailed evaluation of the changes in the milk parameters due to parity, time of evaluation, and farm are summarized in Table 5. In the case of parity, it is possible to observe a decreasing trend in protein (3.87%; 3.67%; 3.57%) and casein (3.09%; 2.93%; 2.79%) content with increasing parity, which is accompanied by a statistically significant (*p* < 0.05) decrease in SNF. The opposite dynamic was observed for FP. The values for total protein content at dry-off and after calving were 3.81% and 3.60%, respectively, a difference of −0.21% (*p* < 0.05). Other milk parameters were not affected by the different stages of lactation. There were also differences among farms for fat, casein, lactose, SNF, TS, and FP. The second farm showed the highest percentage of fat (4.35%), which was 1.59% and 1.67% (*p* < 0.05), more than the first and third farms, respectively. These results match with values of TS, and farm number two showed the highest content of TS (13.43%, *p* < 0.05). Finally, yet importantly, it is necessary to mention the values of SCC. As seen from Table 5, no effect significantly affects SCC, which is consistent with Table 4. Although significant effects on SCC were not reached, large differences (up to 212,000 cells per mL) between farms are observed, which is the subject of further detailed descriptive statistics.

More detailed statistics were compiled for SCC due to the previous non-significant results reported above. According to the arithmetic mean SCC values for each farm, it is clear that the dry-off and initiation of milk secretion management vary at the herd level. An extreme change in SCC values was recorded by Farm 1. At dry-off, the mean SCC value of quarter milk samples was 188,316 cells per mL. After calving, there was an increase of 448,434 to 636,750 cells per mL. If the SCC value of 200,000 cells per mL is considered an indicator of bacterial infection [18], 30% of the quarters at Farm 1 exceeded this threshold. After calving, this proportion decreased to 27%. The arithmetic mean SCC value at dry-off on Farm 2 was 189,303 cells per mL, with the proportion of quarters above the 200,000 cells per mL threshold of 42%. At the beginning of lactation, Farm 2 experienced a large drop in the mean SCC and percentage of quarters above the threshold. The arithmetic mean SCC was 94,966 cells per mL, and the percentage of quarters above the threshold for bacterial infection was 16%. The arithmetic mean SCC values at dry-off and after calving on Farm 3 were 153,389 and 254,622 cells per mL, respectively. The proportion of quarters above the SCC threshold of 200,000 cells per mL was 24% at dry-off on Farm 3. After calving, this proportion decreased to 18%.

## 4. Discussion

The dry period is considered a critical period during lactation, the management of which is crucial for maintaining good milk yield, cow health, and efficient reproductive performance in the following lactation. Globally, the dry period is defined as a not-lactating state, but there is no standardized procedure for an initiation, even though it is a routine practice in dairy herds. Methods of stopping milk production vary across countries and herds, and sometimes even between cows. This is probably due to differences between dairy farms [19]. These differences were evident from our results.

When we compared the average values of the cashable milk components, such as fat and protein, with different studies [20,21,22] that have recorded the content of these components in an early and late stage of lactation, we observed that, for example, the fat content of our samples was lower. We measured 3.36% milk fat before dry-off and 2.96% after calving. Even though it is typical that fat content increases towards the end of lactation as milk yield decreases, it generally does not reach as high values as at the beginning of lactation. Multiple factors could explain the lower fat content at both times of evaluation. One of these factors may be different milk collection dates throughout the year because fat is the most variable component of milk during the season [23], which was confirmed by the regression included in the statistical model. The high differences between the minimum and maximum, the relatively high coefficient of variation, and the GLM procedure confirmed that the milk fat varied according to the farms included in the experiment. As Holstein cows were reared on all farms, the variability in milk fat content could be attributed to, for example, genetic background, as confirmed by Liu et al. [24] in their genome-wide association study for milk production and quality traits in Holstein cattle. Acidosis, the most important rumen dysfunction in dairy cows associated with low milk fat syndrome, may be to blame [25,26]. The lower fat content may also be affected by the occurrence of intramammary infections that arise from metabolic diseases (e.g., ketosis, hypocalcemia) or due to other internal and external factors that have compromised the health of the cow not only after calving but also in the past (e.g., improper dry cow management or diet composition) [27,28,29,30,31].

In contrast, the mean protein content at dry-off slightly exceeds or equals the values reported in other studies [20,21,22], which is consistent with decreasing milk yield. The total protein dropped slightly after calving to a value considered appropriate for this period, given that milk samples were taken on the sixth day in milk when colostrum with high whey protein content was no longer secreted. A deficient milk fat-to-protein ratio is observed when evaluating the relationship between average fat and protein content after calving (<1). A score between 1.0 and 1.5 has been considered normal for Holstein cows in early lactation [32]. The meager fat-to-protein ratio at Farm 1 (0.5) and Farm 3 (0.6) may indicate rumen pH problems accompanied by milk fat depression. These symptoms may indicate those above subacute ruminal acidosis caused by poor acid-base regulation in the foregut [33,34]. The casein content decreased with the decrease in total protein, as confirmed by the strong correlations. Both contents were also affected by parity. Cows produced the lowest total protein and casein content in the third and higher lactations. Yang et al. [35] showed a similar trend and observed a conclusive decrease in total protein in the fourth lactation compared to the second. Additionally, Bonfatti et al. [36] confirmed the decreasing casein content with increasing parity in buffaloes.

The lactose content at dry-off and after calving was around 5%, which can be considered a typical value for this milk component [37]. No significant results between the different evaluation times of milk were established, which corresponds to the fact that the lactose does not follow the standard lactation curve, unlike the fat and protein content [38]. The negative fair to moderately strong correlation between fat and lactose content at dry-off and after calving can be attributed to the fact that lactose determines the amount of absorbed water in the alveoli, implying that lactose content is positively related to the milk volume, in contrast to fat content, which increases with decreasing volume [39]. A fair correlation between total protein and lactose was confirmed after calving. Our experiment did not confirm the gradual decrease in lactose content across parities reported in another publication [38]. There was a significant decrease between the second and third and more lactations, which can be explained by a higher incidence of mastitis in older cows, which is also supported by our results for SCC [40]. Lower lactose content in primiparous cows may be due to the inappropriate rearing of heifers [41] or could also be disturbed by clinical mastitis in primiparous cows, which is more frequent in heifers than in cows in the first days after calving [42], as indicated by the SCC value. The negative effect of SCC on lactose content is discussed below.

As expected, the SNF and TS content correlated with the trend of the leading milk components. The freezing point is inconsistent with the standard range, and the value increased to −0.507 °C after calving [43]. Although the freezing point value was higher, there was no water adulteration. Other factors can explain this value. An example is the effect of the stage of lactation. A lower freezing point was recorded at the end of lactation than at the beginning. A similar lactation effect has been described by Henno et al. [44] and by experts in the Czech Republic [43]. This variation over time is consistent with changes in milk components. The freezing point is strongly negatively correlated with SNF content at both times of evaluation, which corresponds to the fact that the depression of the freezing point is mainly related to the lactose content [43,45]. Our results also show that the aging of dairy cows is associated with an increase in freezing point; inter-farm variation was also confirmed, which is in line with published literature mentioning many genetic and non-genetic factors acting on this parameter [44,46]. The average titratable acidity values before and after calving were within the recommended range for raw milk (6.2 to 7.8 °SH) [47]. After calving, a decrease in titratable acidity was observed, which is consistent with the changes occurring during early lactation. According to the reasonable positive correlation obtained between titratable acidity and total protein, including casein content, it is clear that the acidic groups of casein contribute significantly to acidity [48]. The fact that protein fundamentally influences titratable acidity has been confirmed by other literature [48,49]. Titratable acidity also decreased with the age of cows, which is in line with the same trend for protein.

Following the new Regulation of European Union 2019/6 [5] restrictions, it is necessary to establish tools to refrain from the preventive administration of antibiotics during the dry period while maintaining a certain level of herd health. One option that could be used to manage selective dry cow therapy is SCC. For this reason, SCC values from three different farms were investigated in this study. The average SCC values obtained before and after calving (170,659 and 254,637 cells per mL) can be considered compliant according to Regulation 2004/853 [4]. However, the different SCC between the minimum and maximum and the very high coefficients of variation indicate significant incongruities in the proportions of cows with intramammary infections between herds. When we look at milk’s main components and properties, we find that SCC correlated significantly only with lactose content, which is the body’s corresponding response to damage to secretory cells by inflammation and infection [38]. An identical negative correlation was also reported by Cinar et al. [50].

Furthermore, there was no significant effect of primiparous and multiparous cows, time of evaluation of milk samples, farm, or the day of calving in SCC. Although the GLM procedure did not confirm the significant effect of the farm on SCC, a close look at these values for individual farms shows considerable differences. At dry-off, the highest average was achieved at Farm 2, corresponding to the highest proportion of quarters with SCC above 200,000 cells per mL. However, the proportion of quarters exceeding values indicating bacterial infection decreased to 16% after calving. This decline may be the result of proper management of the dry period. The SCC value obtained after calving on Farm 1 is alarming (636,750 cells per mL) as it indicates that the incidence of intramammary mastitis persisted during the dry period, even though blanket dry cow therapy was implemented on all three farms. The persistent proportion of quarters with increased SCC after calving (27%) on Farm 1 may be attributed to commonly observed increases in SCC during the immediate post-partum period, even for uninfected quarters [51], or may be related to the low fat-to-protein ratio mentioned above. Whether the cause of the problem is an inadequate composition of the diet or contagious or environmental mastitis pathogens would need to be verified by further analysis. The high SCC could also be supported by developing a new infection during the dry period following poor teat-end integrity or the absence of a keratin plug [52]. Even though the average SCC values on Farm 1 at the start of the dry period are below the threshold indicative of bacterial infection (200,000 cells per mL; [18]), post-calving SCC values show the absence of mastitis prevention, without which selective dry cow therapy is difficult to apply. The identical situation with the SCC value is at dry-off on Farm 2. We would only recommend particular dry cow therapy to Farm 3. This is due to the lowest proportion of quarters above 200,000 cells per mL at dry-off with a decreasing tendency after calving. In contrast, the other two farms must improve their cows’ udder health and mastitis control programs before they might consider switching to selective dry cow treatment.

Finally, SCC thresholds for managing selective dry cow therapy were taken from two research studies and applied to the separation of analyzed quarter milk samples. According to the threshold values recommended by Zecconi et al. [53], 72% of the quarters would have been dried off without the administration of antibiotics (primiparous cows < 100,000 cells per mL and multiparous cows < 200,000 cells per mL). The higher SCC thresholds were taken from Scherpenzeel et al. [54]. They opted for non-antibiotic therapy if the SCC was no higher than 150,000 cells per mL for cows on their first lactation and 250,000 cells per mL for cows on their second or higher lactation. Using these thresholds, 75% of the quarters would be selectively dried-off. However, we believe it could be beneficial for maintaining good udder health on the farm to only aim for non-antibiotic dry-off for 20% of the herd when switching to selective dry cow treatment for the first time. It should be noted that this monitoring is a pilot basis for developing the methodology for selective dry cow treatment based on SCC in the Czech Republic. To establish the final SCC thresholds for Czech dairy farms, other factors influencing the choice of therapy related to maintaining the efficiency and health of the herds will have to be considered.

## 5. Conclusions

Our findings suggest that more parameters that affect milk quarter samples need to be considered for the proper selection of cows for selective dry cow therapy. One of them is somatic cell count as the main health indicator. The lactose content as the main component of milk was significantly correlated with this indicator. It should also be considered the fat and protein content, which may be related to metabolic diseases. These results will be used to develop a methodology for the Czech dairy farms to assist in implementing selective dry cow therapy. This methodology would facilitate the selection of dairy cows for antibiotic drying-off using a flow chart based on milk parameters available to the farmers.

## Figures and Tables

**Table 1 animals-13-00712-t001:** Descriptive statistics of analyzed milk parameters before the dry period and after calving, including all quarter milk samples in the experiment.

Time	Variable	n	$\bar{x}$	Minimum	Maximum	SE	CV (%)
At dry-off	Fat (%)	193	3.36	1.25	7.69	0.09	37.57
	Protein (%)	193	3.98	2.86	5.30	0.03	11.00
	Casein (%)	193	3.15	1.09	4.37	0.03	14.07
	Lactose (%)	193	4.82	2.89	5.59	0.04	10.65
	SNF content (%)	193	9.58	7.09	10.94	0.05	6.95
	TS content (%)	193	12.92	10.30	17.00	0.09	10.22
	FP (°C)	193	−0.551	−0.671	−0.400	0.00	7.07
	Titratable acidity (°SH)	209	7.04	2.82	11.35	0.09	18.52
	SCC (cells/mL)	211	170,659	1500	2,255,000	20,250.45	172.36
After calving	Fat (%)	154	2.96	1.00	10.85	0.14	60.29
	Protein (%)	175	3.35	2.70	4.18	0.02	8.80
	Casein (%)	175	2.65	1.59	3.24	0.02	9.62
	Lactose (%)	175	5.05	2.80	5.61	0.03	8.01
	SNF content (%)	175	9.20	6.74	9.95	0.03	4.83
	TS content (%)	158	12.07	8.88	18.55	0.13	13.30
	FP (°C)	175	−0.507	−0.547	−0.350	0.00	4.74
	Titratable acidity (°SH)	174	6.94	2.40	9.89	0.08	15.36
	SCC (cells/mL)	175	254,637	3000	7,600,000	69,784.63	362.54

$\bar{x}$—arithmetic means; SE—standard error of arithmetic means; CV—coefficient of variation; SNF content—solids-not-fat; TS—total solids; FP—freezing point; SCC—somatic cell count.

**Table 2 animals-13-00712-t002:** Pearson’s correlation coefficients between the analyzed milk parameters supplemented with the significance level. The dataset contains quarter milk samples analyzed before the dry period. The correlation coefficients were evaluated by [17] as follows: less than 0.30 (−0.30) poor; 0.30 to 0.59 (−0.30 to −0.59) fair; 0.60 to 0.79 (−0.60 to −0.79) moderately strong; at least 0.8 (−0.8) very strong.

		Fat	Protein	Casein	Lactose	SNF Content	TS Content	FP	Titratable Acidity	SCC
Parity	r	−0.154	−0.239	−0.296	−0.107	−0.271	−0.283	0.282	−0.036	0.076
	*p*-value	0.033	<0.01	<0.01	0.138	<0.01	<0.01	<0.01	0.606	0.270
	n	193	193	193	193	193	193	193	209	211
Fat	r		0.230	0.123	−0.405	−0.093	0.879	−0.163	−0.079	0.167
	*p*-value		<0.01	0.088	<0.01	0.198	<0.01	0.024	0.275	0.02
	n		193	193	193	193	193	193	193	193
Protein	r			0.919	−0.070	0.671	0.560	−0.527	0.247	0.018
	*p*-value			<0.01	0.335	<0.01	<0.01	<0.01	<0.01	0.807
	n			193	193	193	193	193	193	193
Casein	r				0.296	0.879	0.557	−0.689	0.327	−0.100
	*p*-value				<0.01	<0.01	<0.01	<0.01	<0.01	0.167
	n				193	193	193	193	193	193
Lactose	r					0.678	−0.080	−0.556	0.301	−0.355
	*p*-value					<0.01	0.270	<0.01	<0.01	<0.01
	n					193	193	193	193	193
SNF content	r						0.381	−0.782	0.448	−0.241
	*p*-value						<0.01	<0.01	<0.01	<0.01
	n						193	193	193	193
TS content	r							−0.521	0.096	0.058
	*p*-value							<0.01	0.183	0.422
	n							193	193	193
FP	r								−0.140	0.145
	*p*-value								0.052	0.045
	n								193	193
Titratable acidity	r									−0.289
	*p*-value									<0.01
	n									209

SNF content—solids-not-fat; TS—total solids; FP—freezing point; SCC—somatic cell count.

**Table 3 animals-13-00712-t003:** Pearson’s correlation coefficients between the analyzed milk parameters supplemented with the significance level. The dataset contains milk quarter samples analyzed after calving. The correlation coefficients were evaluated by [17] as follows: less than 0.30 (−0.30), poor; 0.30 to 0.59 (−0.30 to −0.59), fair; 0.60 to 0.79 (−0.60 to −0.79), moderately strong; and at least 0.8 (−0.8), very strong.

		Fat	Protein	Casein	Lactose	SNF Content	TS Content	FP	Titratable Acidity	SCC
Parity	r	0.184	−0.147	−0.127	−0.089	−0.192	0.101	0.241	−0.205	0.023
	*p*-value	0.023	0.052	0.095	0.239	0.011	0.205	<0.01	<0.01	0.765
	n	154	175	175	175	175	158	175	174	175
Fat	r		0.136	0.117	−0.630	−0.403	0.975	0.247	−0.053	0.012
	*p*-value		0.092	0.149	<0.01	<0.01	<0.01	<0.01	0.515	0.880
	n		154	154	154	154	154	154	153	154
Protein	r			0.903	−0.301	0.590	0.248	−0.430	0.385	0.021
	*p*-value			<0.01	<0.01	<0.01	<0.01	<0.01	<0.01	0.787
	n			175	175	175	158	175	174	175
Casein	r				0.038	0.768	0.253	−0.620	0.415	−0.137
	*p*-value				0.621	<0.01	0.01	<0.01	<0.01	0.071
	n				175	175	158	175	174	175
Lactose	r					0.552	−0.560	−0.533	0.106	−0.277
	*p*-value					<0.01	<0.01	<0.01	0.162	<0.01
	n					175	158	175	174	175
SNF content	r						−0.243	−0.846	0.461	−0.241
	*p*-value						<0.01	<0.01	<0.01	0.01
	n						158	175	174	175
TS content	r							0.065	0.043	−0.001
	*p*-value							0.418	0.597	0.993
	n							158	157	158
FP	r								−0.340	0.099
	*p*-value								<0.01	0.192
	n								174	175
Titratable acidity	r									−0.188
	*p*-value									0.013
	n									174

SNF content—solids-not-fat; TS—total solids; FP—freezing point; SCC—somatic cell count.

**Table 4 animals-13-00712-t004:** The descriptive statistics for GLM evaluation, which describes the significance of the model and the effects in the model equation for analyzed milk parameters in quarter samples.

	Model		Parity		Time of Evaluation		Farm		Regression on the Day of Calving	
	r^2^	*p*-Value	F-Test	*p*-Value	F-Test	*p*-Value	F-Test	*p*-Value	F-Test	*p*-Value
Fat	0.279	<0.01	0.14	0.868	2.14	0.145	54.94	<0.01	17.98	<0.01
Protein	0.481	<0.01	14.78	<0.01	5.99	0.015	1.84	0.16	18.43	<0.01
Casein	0.405	<0.01	17.34	<0.01	2.53	0.113	3.64	0.027	13.19	<0.01
Lactose	0.143	<0.01	2.89	0.057	0.35	0.552	13.83	<0.01	5.9	0.016
SNF content	0.186	<0.01	11.98	<0.01	0.93	0.337	9.06	<0.01	0.62	0.431
TS content	0.266	<0.01	3.77	0.024	0.52	0.473	29.53	<0.01	17.23	<0.01
FP	0.389	<0.01	12.07	<0.01	4.93	0.027	3.61	0.028	11.34	<0.01
Titratable acidity	0.035	0.038	3.85	0.022	1.57	0.211	1.97	0.144	1.59	0.208
SCC	0.027	0.115	1.35	0.259	0.27	0.604	2.45	0.088	0	0.987

r^2^—determination coefficient; *p*-value—significance of effects; SNF content—solids-not-fat; TS—total solids; FP—freezing point; SCC—somatic cell count.

**Table 5 animals-13-00712-t005:** Significant differences for monitored effects, least squares means, and their standard errors for milk parameters from analyzed quarter samples.

Effect	Level	Fat (%)	Protein (%)	Casein (%)	Lactose (%)	SNF Content (%)	TS Content (%)	FP (^o^C)	Titratable Acidity (^o^SH)	SCC (Cells/mL)
		LSM ± SE	LSM ± SE	LSM ± SE	LSM ± SE	LSM ± SE	LSM ± SE	LSM ± SE	LSM ± SE	LSM ± SE
Parity	First	3.31 ± 0.175	3.87 ± 0.046 ^a^	3.09 ± 0.045 ^a^	4.90 ± 0.058	9.59 ± 0.071 ^a^	12.82 ± 0.175 ^a^	−0.54 ± 0.004 ^a^	7.21 ± 0.151 ^a^	191,069 ± 82,919
	Second	3.20 ± 0.138	3.67 ± 0.035 ^b^	2.93 ± 0.034 ^b^	4.99 ± 0.044 ^a^	9.46 ± 0.054 ^a^	12.67 ± 0.135	−0.53 ± 0.003 ^a^	7.11 ± 0.116	158,086 ± 63.407
	Third and more	3.27 ± 0.110	3.57 ± 0.029 ^b^	2.79 ± 0.028 ^c^	4.85 ± 0.036 ^b^	9.21 ± 0.044 ^b^	12.32 ± 0.110 ^b^	−0.52 ± 0.002 ^b^	6.79 ± 0.092 ^b^	285,624 ± 50,345
Time of evaluation	At dry-off	3.02 ± 0.148	3.81 ± 0.041 ^a^	3.00 ± 0.040	4.88 ± 0.052	9.48 ± 0.063	12.49 ± 0.149	−0.54 ± 0.004	6.86 ± 0.132	171,983 ± 72,245
	After calving	3.50 ± 0.215	3.60 ± 0.054 ^b^	2.87 ± 0.052	4.94 ± 0.067	9.36 ± 0.082	12.72 ± 0.213	−0.52 ± 0.005	7.21 ± 0.179	251,203 ± 97,922
Farm	1	2.76 ± 0.191 ^a^	3.70 ± 0.047	2.90 ± 0.045	4.92 ± 0.059	9.41 ± 0.072	12.09 ± 0.190 ^a^	−0.52 ± 0.004 ^a^	7.00 ± 0.142	331,511 ± 77,882
	2	4.35 ± 0.130 ^b^	3.66 ± 0.034	2.90 ± 0.033 ^a^	4.76 ± 0.043 ^a^	9.28 ± 0.052 ^a^	13.43 ± 0.129 ^b^	−0.53 ± 0.003	6.92 ± 0.115	119,344 ± 63,249
	3	2.68 ± 0.105 ^a^	3.74 ± 0.028	3.00 ± 0.027 ^b^	5.05 ± 0.035 ^b^	9.56 ± 0.043 ^b^	12.29 ± 0.103 ^a^	−0.54 ± 0.002 ^b^	7.20 ± 0.093	183,924 ± 50,834

SNF content—solids-not-fat; TS—total solids; FP—freezing point; SCC—somatic cell count; LSM—least squares means; SE—standard errors; ^a, b, c^ statistical significance *p* <0.05.

## Data Availability

The data presented in this study are available on request from the corresponding author.

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
