# Peer review of "Dynamics of Milk Parameters of Quarter Samples before and after the Dry Period on Czech Farms"

_animals, 2023, doi:10.3390/ani13040712_

Round 1
Reviewer 1 Report
Major concerns:
a) In the title “in relation to antibiotic application” and the objective “This study aimed to examine whether Czech dairy farms can proceed to antibiotic-free dry cow therapy based on the tested parameters”.
ü The study methods and results did not address the study objective as there was no mention for the antibiotic application in the methods. How did the authors address the antibiotic application in their study?
b) Knowledge gap is not clear
c) The study objective is vague. Please be more specific in your objectives.
d) There is no study hypothesis.
Materials and methods:
1- Sampling:
a) The number of samples are not consistent in the whole manuscript. For example:
1. line 81 “374 quarter milk samples”
2. line 85 “Ninety-five dairy cows were included” if 95 cows were enrolled the number of milk samples should be [95(cows)*4(quarters)*2(sampling points)] which equals 760 samples.
3. Tables 1,2,3 have different numbers for each component
4. Table 6: has 386 samples.
· Did the authors samples all the enrolled cows?
· Did the authors test all the milk samples from all enrolled cows?
· Did the analysis include all the milk samples?
ü If yes, please clarify why there are big variations in the numbers?
ü If no, please clarify why not? And how that was explained and justified in the manuscript? as that may affect the validity and generalizability of the results.
a) In the title of the study “Reduction of Antibiotic Application” but there is no description for the antibiotic applications in the methods. How did the authors evaluate the effect of monitoring of SCC and milk components on the reduction of antibiotic application in their study?
b) What was the inclusion and exclusion criteria for the enrolled herds?
c) What was the inclusion and exclusion criteria for the enrolled cows?
b) If the average number of cows per herd 100-400 cows why the study period was over 2 years to sample 95 cows from 3 herds?
c) Were any of these cows sampled in two successive lactations? As the study period was 2 years.
d) Who did the enrollment and sampling of the enrolled cows?
e) Did the authors measure the milk production for each cow? How they account for the effect of milk production on the milk components in the analysis?
f) There is no description of the number of cows in each parity and the length of the dry period of the enrolled cows which may affect the study outcomes.
g) What was the unit of the analysis? Is it quarter or cow?
h) How did the authors account for the repeated measures (4 quarters for each cow * two times sampling = 8 observations from each cow)?
Results:
a) Tables 2 and 3 I would recommend substituting with a table as a supplement if someone wants to see more details, but it did not add much and hard to follow in their current format.
b) For the models tables (4, 5) it would be better to present the magnitude of the effect of each variable (coefficient) rather than the least square means.
c) -What regression on day of calving mean (model equation line 130, table 4)?
d) -Table 6:
e) Does the (n) numbers in the table represent quarters or cows?
f) Why there are big variation in the (n) between beginning of dry period and after calving?
Minor comments:
a) The title is very long and wordy and more general. I would recommend shortening the title and be more specific.
b) Line 15: I would recommend adding the word Antimicrobial before the word Resistance.
c) Line 18: “selected milk parameters” What are they? Please, add them to the text.
d) Line 19: “during the interruption of lactation of cows” what does that mean? Please clarify in the whole manuscript.
e) Lines 21-21: “antibiotics are not applied prudently” Please describe how? For example was that due to wrong protocols, incorrect dose, duration, extra label use,……………….
f) “Significant changes in the mastitis prevention management of individual farms are needed” what are they?
g) Line 25: “milk samples collected from each teat” Do you mean each quarter? Please clarify in the whole manuscript.
h) Line 26: “at the beginning of the dry period” I would recommend changing to “at dry off” in the whole manuscript.
i) Line 27: “day of calving were evaluated before and after the dry period” What evaluation of day of calving before dry period mean? How did the authors evaluate day of calving before dry period?
j) Lines 27-28 “The values of some of the leading milk components pointed not only to possible metabolic diseases but also to inappropriate rearing practices or inadequate diet composition.” This statement has nothing to do with the study objectives and the meaning is not clear. Please clarify what are these milk components and how they pointed out to metabolic diseases, inappropriate rearing practices, or inadequate diet composition?
k) Lines 29-30: “somatic cell counts on one farm exceeded 600,000 cells per mL after calving” was this value an average for all cows? Or every cow on the dairy exceeded 600,000 cells/ml.
l) Lines 30-31: “According to the results of our monitoring,” what are these results?
m) Lines 32-33: “It should be recognized that antibiotic overuse is a global problem that needs to be addressed in all relevant sectors, including dairy farms.” This conclusion is beyond the study objective. I would recommend the conclusion to be more specific to the study findings.
n) Lines 59-60: “The above shows that selective dry cow treatment is scarcely applied in the Czech Republic” I am not sure what the authors refer to when they said the above shows? as there is no data about the percentage of dairies using selective dry cow therapy.
o) Line: “cows are exposed to many stressors” Please mention those stressors.
Author Response
Dear Reviewer,
We are very pleased to have been given the opportunity to revise our manuscript. We also appreciate the time and effort you have dedicated to providing insightful feedback on ways to strengthen our paper. Our responses to your comments are mentioned line-by-line in red text in Word.
Major concerns:
- a)In the title “in relation to antibiotic application” and the objective “This study aimed to examine whether Czech dairy farms can proceed to antibiotic-free dry cow therapy based on the tested parameters”.
ü The study methods and results did not address the study objective as there was no mention for the antibiotic application in the methods. How did the authors address the antibiotic application in their study?
Thank you for your comment. In our experiment, we did not manage the application of antibiotics, but we mentioned several times in our manuscript that all farms used the blanket dry cow therapy. Following this, we evaluated the somatic cells count and other milk parameters to highlight the fact that selective therapy is not being implemented but could be. We have decided to rephrase the title of the manuscript to: “Dynamics of Milk Parameters of Quarter Samples Before and After the Dry Period on the Czech Farms”. Based on your comment, we have also reformulated the aim to be consistent with the methods and results. We have changed sentences in L: 102 “As the quality of milk in the Czech Republic is increasing steadily in terms of hygiene and health indicators (composition, count of microorganisms, SCC, residues of inhibiting substances), we hypothesized that selective dry cow therapy could be applied to part of the teats. To investigate this, the aims of this study were to measure SCC and other milk parameters before and after dry period at the level of individual teats; assess the correlation between these values and explore the influence of parity, farms, time of evaluation, and day of calving were also examined to better understand possible variations in parameters.”
- b) Knowledge gap is not clear
Our experiment is a pilot study to investigate the possibility of selective dry cow therapy to reduce the use of intramammary antibiotics on the Czech dairy farms. We are aware that the manuscript does not present any particulars new findings, however, we believe it provides an interesting comparison with foreign literature and draws attention to the issue of antibiotic overuse.
- c) The study objective is vague. Please be more specific in your objectives.
The study objective has been reformulated in L: 79.
- d) There is no study hypothesis.
Thank you for the notice. We have included the study hypothesis in L: 79
Materials and methods:
1- Sampling:
- a) The number of samples are not consistent in the whole manuscript. For example:
- line 81 “374 quarter milk samples”
- line 85 “Ninety-five dairy cows were included” if 95 cows were enrolled the number of milk samples should be [95(cows)*4(quarters)*2(sampling points)] which equals 760 samples.
- Tables 1,2,3 have different numbers for each component
- Table 6: has 386 samples.
Thank you for pointing out this problem, and we apologize that we did not notice this glaring mistake before submission. We rechecked the numbers in the whole manuscript and found few inconsistencies between statistical analysis and numbers presented in manuscript. The numbers were corrected and are now in full correspondence with statistical analysis. We added detail explanation about numbers of tested milk samples at the beginning of materials and methodology as previously stated numbers were confusing. Tables 1, 2, 3 have slightly different numbers for some components, as the results for those analysis were invalid and not counted for the statistical evaluation. Changes have been recorded in L: 115 “In total, 386 quarter milk samples were collected and evaluated from 3 dairy farms (Farm 1 = 84 samples; Farm 2 = 114 samples; Farm 3 = 188 samples) in the Czech Republic from December 2021 to June 2022. The samples were collected in two distinct periods – at the dry off (n = 211) and after calving (n = 175), while one cow had one non-functional teat and was only milked from 3 teats. The distribution of cows based on lactation number corresponded to the profile of the herds, when 21 cows entered the test at the first lactation, 14 cows at the second lactation and 18 cows at the third and higher lactation. Some of the cows sampled at the dry off were culled during dry period and few were culled due post-partum health problems (n = 9).“ Furthermore, Table 6 were deleted from the manuscript on the recommendation of another reviewer as they raised meaningful objections for evaluating SCC as arithmetic means, which we acknowledged. However, we reworked the paragraph belonging the table and added more results about the situation of SCC of quarter milk samples after dry-off and after calving on farms.
- Did the authors samples all the enrolled cows? Yes, we did.
- Did the authors test all the milk samples from all enrolled cows? Yes, we did.
- Did the analysis include all the milk samples? Yes, we did, however, results for some parameters in few samples were invalid - outliers.
ü If yes, please clarify why there are big variations in the numbers?
The explanation is given above.
ü If no, please clarify why not? And how that was explained and justified in the manuscript? as that may affect the validity and generalizability of the results.
- a) In the title of the study “Reduction of Antibiotic Application” but there is no description for the antibiotic applications in the methods. How did the authors evaluate the effect of monitoring of SCC and milk components on the reduction of antibiotic application in their study?
Thank you for pointing. The information about antibiotic application has been added in the method in L: 130 “The blanket dry cow therapy was the standard way to dry-off cow on all of farms”. As we mentioned above, our aim was to evaluate the possibilities of application of selective dry cow therapy. To investigate this, the aim of this study was to measure SCC and other milk parameters. The next step of our research will be the expansion of the dataset and subsequent testing of different methodologies for selective dry cow therapy with aim of reduction of antibiotic application.
- b) What was the inclusion and exclusion criteria for the enrolled herds?
As we mention in the paragraph “Material and Methods”, herds were selected according to the typical breeding profile in the Czech Republic. We also considered the size of the herd to represent the average herd size in our conditions. In the “Introduction” we also mentioned that in 2019 more than 80% of dairy cows were dried-off by intramammary antibiotics, for this reason we included in the experiment dairy farms where only blanket dry cow therapy was applied. In addition, the farms were located in the regions that are typical of the climatic and geographical conditions in the Czech Republic.
- c) What was the inclusion and exclusion criteria for the enrolled cows?
No exclusion criteria have been set. Quarter milk samples were collected from all cows scheduled for a dry period.
- b) If the average number of cows per herd 100-400 cows why the study period was over 2 years to sample 95 cows from 3 herds?
Thank you for pointing. The description of the sampling timeline in the Material and Methodology was not accurate. The milk samples were taken from December 2021 to June 2022, this information has been written in L: 115 “In total, 386 quarter milk samples were collected and evaluated from 3 dairy farms (Farm 1 = 84 samples; Farm 2 = 114 samples; Farm 3 = 188 samples) in the Czech Republic from December 2021 to June 2022.”
- c) Were any of these cows sampled in two successive lactations? As the study period was 2 years.
Thank you for pointing. As we mentioned above, we have not collected samples during two lactations. However, we had re-checked the dataset of enrolled cows, and there were no duplicates (no cow was sampled in two consecutive lactations).
- d) Who did the enrollment and sampling of the enrolled cows?
Members of the author's team.
- e) Did the authors measure the milk production for each cow? How they account for the effect of milk production on the milk components in the analysis?
Thank you for pointing. We did not record milk production, we are aware that some authors report confirmation of correlation e.g. between milk production and fat content or protein content (Friggens and Rasmussen, 2001; Løvendahl and Chagunda, 2011). However, we believe that the inclusion of this effect on the milk components is more appropriate for studies testing different milking frequencies or different lengths of dry periods than for our experiment. But even so, this effect is commented on, we have mentioned several times in the Discussion in L: 322, 357, 363.
- f) There is no description of the number of cows in each parity and the length of the dry period of the enrolled cows which may affect the study outcomes.
Thank you for the recommendation. A note on the number of dairy cows in different parities has been added to the Material and Method in L: 119 “The distribution of cows based on lactation number corresponded to the profile of the herds, when 21 cows entered the test at the first lactation, 14 cows at the second lactation and 18 cows at the third and higher lactation. Some of the cows sampled at the dry off were culled during dry period and few were culled due post-partum health problems (n = 9).” The length of the dry period was also mentioned in L: 131 “The length of the dry period was 60 d for all quarters of all cows.”
- g) What was the unit of the analysis? Is it quarter or cow?
Thank you for pointing. It was a quarter of the udder. To make this unit obvious, we have added it to certain parts of the manuscript (e.g. table headings in L: 204, 224, 204, 242, 255).
- h) How did the authors account for the repeated measures (4 quarters for each cow * two times sampling = 8 observations from each cow)?
Thank you for this question. Quarters were evaluated individually. Even though including random effect of the animal would make sense for this kind of evaluation, in the end we chose not to include it as we wanted to focus on researching selective dry cow therapy for individual quarters, and therefore we considered them individually.
Results:
- a) Tables 2 and 3 I would recommend substituting with a table as a supplement if someone wants to see more details, but it did not add much and hard to follow in their current format.
Thank you for recommendation. We understand that the current format is not very comfortable, but Journal Animals provides an online view with pop-ups for tables, so we decided to keep the tables in the manuscript.
- b) For the models tables (4, 5) it would be better to present the magnitude of the effect of each variable (coefficient) rather than the least square means.
Thanks for the suggestion. In Table 4 we state the significance of individual effects and the determination coefficient of the model equation. We chose effects into the model equation based on STEPWISE method of REG procedure. Based on this principle, the model equation contains only effects that were suitable for the evaluation of most parameters. Therefore, there is no need to do an analysis of the magnitude of the effects. Moreover, most of the effects have already been confirmed as highly influential by number of studies, and as we can see in Table 4 most of them were significant for monitored parameters.
- c) -What regression on day of calving mean (model equation line 130, table 4)?
Day of calving has been included in the model to account for the effect of the season.
- d) -Table 6:
As mentioned above, the table has been removed on the recommendation of another reviewer and replaced with the following text in L: 292 “More detailed statistics were compiled for SCC due to previous non-significant re-sults reported above. According to the arithmetic mean SCC values for each farm, it is clear that the dry-off and initiation of milk secretion management vary at the herd level. The extreme change in SCC values was recorded by Farm 1. At dry-off, the mean SCC value of quarter milk samples was 188,316 cells per mL. After calving, there was an increase of 448,434 to 636,750 cells per mL. If the SCC value of 200,000 cells per mL is considered as an indicator of bacterial infection [18], 30% of the teats at Farm 1 exceeded this thresh-old. After calving, this proportion decreased to 27%. The arithmetic mean SCC value at dry-off on the Farm 2 was 189,303 cells per mL with the proportion of teats above the 200,000 cells per mL threshold of 42%. At the beginning of lactation, Farm 2 experienced a large drop in the mean SCC and percentage of teats above the threshold. The arithmetic mean SCC was 94,966 cells per mL and the percentage of teats above the threshold for bacterial infection was 16%. The arithmetic means SCC values at dry-off and after calving on Farm 3 were 153,389 and 254,622 cells per mL, respectively. The proportion of teats above the SCC threshold of 200,000 cells per mL was 24% at dry-off on Farm 3. After calving, this proportion decrease to 18%.” We hope that now the information is more appropriate.
- e) Does the (n) numbers in the table represent quarters or cows?
- f) Why there are big variation in the (n) between beginning of dry period and after calving?
Minor comments:
- a) The title is very long and wordy and more general. I would recommend shortening the title and be more specific.
Thank you for pointing. The title has been reformulated to “Dynamics of Milk Parameters of Quarter Samples Before and After the Dry Period on the Czech Farms”. We hope that now the title is more specific, and catchy.
- b) Line 15: I would recommend adding the word Antimicrobial before the word Resistance.
The word “Antimicrobial” has been added.
- c) Line 18: “selected milk parameters” What are they? Please, add them to the text.
The selected milk parameters have been added.
- d) Line 19: “during the interruption of lactation of cows” what does that mean? Please clarify in the whole manuscript.
Thank you for pointing. The sentences have been changed in both cases in L: 22 “…during the cessation of milking at the end..” and in L: 87 “During the halting of milk production in dry period…”.
- e) Lines 21-21: “antibiotics are not applied prudently” Please describe how? For example was that due to wrong protocols, incorrect dose, duration, extra label use,……………….
Thank you for pointing. The sentence has been changed in L: 23 “Unfortunately, routine prophylactic antibiotic use was standard to dry-off cow on all farms tested. However, this may change as our results indicate that the introduction of selective dry cow therapy is feasible, but well-known mastitis prevention and control strategies must be in place on farms.”
- f) “Significant changes in the mastitis prevention management of individual farms are needed” what are they?
Thank you for pointing. The sentence has been changed in L: 26 “…, but well-known mastitis prevention and control strategies must be in place on farms”. We refer here to the ten-point prevention and control plan (National Mastitis Council), which we mentioned in the manuscript.
- g) Line 25: “milk samples collected from each teat” Do you mean each quarter? Please clarify in the whole manuscript.
Thank you for pointing. The abstract has been reformulated in L: 29 “This study aimed to monitor milk parameters on three different dairy farms in the Czech Republic to describe the readiness for implementing selective dry cow therapy. Fat, protein, casein, lactose, solids-not-fat content, total solids content, freezing point, titratable acidity, and somatic cells count of quarter milk samples collected from tested Holstein cows were evaluated. Associations between the tested parameters, as well as the effects of parity, farm, day of calving, time of evaluation at dry-off and after calving were assessed. Values of the leading milk components dynamically changed between dry-off and after calving, but only protein content was significantly affected. The most important parameter of our research, the somatic cell count of quarter milk samples, was also not affected by the time of evaluation. Even though a slight increase in mean of somatic cells count during periods of before dry-off and after calving is expected, at dry-off, we observed 30%, 42% and 24% of quarters with somatic cells count above 200,000 cells per mL, while after calving we observed 27%, 16% and 18% of quarters with somatic cells count above 200,000 cells per mL on Farm 1, Farm 2, and Farm 3, respectively. High somatic cell counts (>200 000 cells per mL) are indicative of bacterial infection, as confirmed by the significant negative correlation between this parameter and lactose content. In addition, a deficient milk fat-to-protein ratio was observed on two farms, which may indicate rumen pH problems, as well as the occurrence of intramammary infections. Despite the above, we concluded that according to the thresholds of somatic cell counts for selective dry cow therapy taken from foreign studies, a large part of the udder quarters could be dried-off without the administration of antibiotics, but it is necessary to set up more effective mechanisms for mastitis prevention.”
- h) Line 26: “at the beginning of the dry period” I would recommend changing to “at dry off” in the whole manuscript.
Thank you for recommendation. Changes have been made.
- i) Line 27: “day of calving were evaluated before and after the dry period” What evaluation of day of calving before dry period mean? How did the authors evaluate day of calving before dry period?
The abstract has been reformulated in L: 25
- j) Lines 27-28 “The values of some of the leading milk components pointed not only to possible metabolic diseases but also to inappropriate rearing practices or inadequate diet composition.” This statement has nothing to do with the study objectives and the meaning is not clear. Please clarify what are these milk components and how they pointed out to metabolic diseases, inappropriate rearing practices, or inadequate diet composition?
The abstract has been reformulated in L: 25
- k) Lines 29-30: “somatic cell counts on one farm exceeded 600,000 cells per mL after calving” was this value an average for all cows? Or every cow on the dairy exceeded 600,000 cells/ml.
The sentence has been removed.
- l) Lines 30-31: “According to the results of our monitoring,” what are these results?
Thank you for the comment. The sentence has been removed. We added more results into the abstract.
- m) Lines 32-33: “It should be recognized that antibiotic overuse is a global problem that needs to be addressed in all relevant sectors, including dairy farms.” This conclusion is beyond the study objective. I would recommend the conclusion to be more specific to the study findings.
The abstract has been reformulated in L: 25
- n) Lines 59-60: “The above shows that selective dry cow treatment is scarcely applied in the Czech Republic” I am not sure what the authors refer to when they said the above shows? as there is no data about the percentage of dairies using selective dry cow therapy.
We refer to the sentence mentioned a line above in L: 81 “Indeed, the downward trend in the use of intramammary antibiotics at dry-off stopped, followed by a severe increase of 20% between 2016 and 2019. The data also show that of the 361,430 dairy cows, more than 80% were dried-off by intramammary antibiotics [7]”. This information was taken from a manuscript by authors working at the Institute for State Control of Veterinary Biologicals and Medicines of the Czech Republic (Pokludova, L.; Maxová, L.; Mašková, Z.; Novotná, P.; Chumchalová, J.; Bures, J. Medicinal Products Used in Mastitis Treatment and Prevention – Overview, Trends in Consumption and Imperative on More Prudent Used of Antimicrobials. Veterinarstvi 2021, 71, 82–93). So, we mentioned about the percentage of dairy cows in the Czech Republic are treated with antibiotics at dry-off.
- o) Line: “cows are exposed to many stressors” Please mention those stressors.
The sentence has been reformulated in L: 87 “During the halting of milk production in dry period, there is a significant physiological imbalance during which cows are exposed to several stressors (mammary gland discomfort, social stressors, etc.) that increase susceptibility to intramammary infections [8].”

Reviewer 2 Report
REVIEW
“Evaluation and Monitoring of Somatic Cells Count and Milk 2 Components on Czech Dairy Farms Before and After the Dry 3 Period in Relation to the Reduction of Antibiotic Application”
Recommendation: The manuscript claims extensive reinvestigation before resubmission. My decision is to accept the manuscript only after satisfactory revision.
Title is too long; make it precise and catchy.
Introduction is impressive and smart.
Conclusion needs through modification. It is just the extension of the discussion of your manuscript. Carefully modify this. Give your own perception and future direction of your current research.
English language has to be modified carefully.
Author Response
Dear Reviewer,
We are very pleased to have been given the opportunity to revise our manuscript. We also appreciate the time and effort you have dedicated to providing insightful feedback on ways to strengthen our paper. Our responses to your comments are mentioned line-by-line in red text in Word.
REVIEW
“Evaluation and Monitoring of Somatic Cells Count and Milk 2 Components on Czech Dairy Farms Before and After the Dry 3 Period in Relation to the Reduction of Antibiotic Application”
Recommendation: The manuscript claims extensive reinvestigation before resubmission. My decision is to accept the manuscript only after satisfactory revision.
Title is too long; make it precise and catchy.
Thank you for pointing. The title has been changed “Dynamics of Milk Parameters of Quarter Samples Before and After the Dry Period on the Czech Farms”. We hope that now the title is more specific and catchier.
Introduction is impressive and smart.
Thank you very much for you recognition.
Conclusion needs through modification. It is just the extension of the discussion of your manuscript. Carefully modify this. Give your own perception and future direction of your current research.
Thank you for pointing. The conclusion has been reformulated in L: 474 “It can be concluded that the collection and evaluation of milk quarter samples before and after dry period to obtain data important for the selective dry cow therapy is feasible on the Czech farms, however some farms are better suited for the initiation of selective dry cow therapy than other due to the mastitis prevalence. Our findings suggest that more parameters and influences that affect milk quarter samples need to be considered for the proper selection of cows for selective dry cow therapy. One of them is clearly somatic cell count as the main health indicator. The lactose content as the main component of milk was significantly correlated with this indicator and should also be considered, or the fat and protein content as indicators of metabolic diseases. These results will be used as a basis for the development of a methodology for the Czech dairy farms to assist in the implementation of selective dry cow therapy. The methodology facilitates the selection of dairy cows for antibiotic drying-off using a flow chart based on milk parameters available to the farmers.“
English language has to be modified carefully.
The English language has been checked by an expert. If you want, we can ask for a certificate.

Reviewer 3 Report
General Comments to Authors:
This manuscript is a simple observational study of various milk parameters in a small number of dairy herds in the Czech Republic. The subject matter is likely to be of some interest to some of the readership of this journal. It does not present any particularly novel findings however it does provide some interesting comparisons to available data from other authors in other geographical regions using (presumably) different management systems.
The manuscript is presented quite well. There are some instances of poor choice of words and awkward syntax that detracts from the readability of the manuscript in places. The authors’ meaning is largely apparent to the knowledgeable reader. I think sections of the manuscript are too long and there is scope for the manuscript to be edited to make it more succinct.
I have two major concerns with the study as reported:
1) The authors present a large number of observations of multiple parameters in a relatively small number of animals, in a very small number of herds. There is a lot of emphasis on the correlations between these parameters without providing any indication of why such correlations are of particular relevance to the diagnosis of infections of the mammary gland, nor how such analysis of the data provides advantages in the control of mastitis in herds. It would be far better if the study focussed on parameters that have direct relationships to mammary gland infection without being distracted by analysis of milk components that are largely dependent on other factors, such as nutrition and genetics.
2) The authors do not clearly separate the use of SCC in individual cows (or quarters) from SCC in bulk tank milk. While there two measurements are related, they are used in very different ways for mastitis control. Given that a major aim of the study is to explore the use of SCC as a basis for selective dry cow therapy, this is a significant shortcoming. The authors make several bold statements about the mastitis status in the small number of herds in the study, but provide no information that gives the reader insight into the details of mastitis in these herds. Despite comments such as “SCC obtained after calving on Farm 1, and Farm 3 are alarming…” I could find no information about causative bacteria or environmental challenge in these herds.
Some specific examples:
Table 6
Averaging of individual cow (or individual quarter) SCC is not wise, given the limits of quantification of SCC using the methods described. Precision of SCC counting is very poor at higher SCC. It would be better to describe the data in terms of proportion of cows which fall within categories of SCC.
Is 'n' the number of teats, cows, samples...? What information is available to explain the differences in numbers between the "Beginning of the dry period" and the "After calving" data?
Line 247: Please provide some information about how these were done in the study herds. If you are extrapolating to the situation more broadly in the Czech national herd, please provide some indication of whether the study herds were representative. Are there any studies that describe how frequently various management approaches of relevance to mastitis, and specific mastitis control strategies, are implemented within Czech herds?
Line 340: Blanket DCT is not a guarantee that bulk tank SCC will decline from one lactation to the next. These assertions need to recognise the multifaceted nature of mastitis infection within herds. This study seems to be drawing conclusions based on 3 herds with a paucity of information about mastitis control within these few herds.
Line 357: Ignores overlying recommendations based on herd BMCC
Line 371: How do you determine/define "prudently?
Author Response
Dear Reviewer,
We are very pleased to have been given the opportunity to revise our manuscript. We also appreciate the time and effort you have dedicated to providing insightful feedback on ways to strengthen our paper. Our responses to your comments are mentioned line-by-line in red text in Word.
General Comments to Authors:
This manuscript is a simple observational study of various milk parameters in a small number of dairy herds in the Czech Republic. The subject matter is likely to be of some interest to some of the readership of this journal. It does not present any particularly novel findings however it does provide some interesting comparisons to available data from other authors in other geographical regions using (presumably) different management systems.
The manuscript is presented quite well. There are some instances of poor choice of words and awkward syntax that detracts from the readability of the manuscript in places. The authors’ meaning is largely apparent to the knowledgeable reader. I think sections of the manuscript are too long and there is scope for the manuscript to be edited to make it more succinct.
Thank you for your feedback.
I have two major concerns with the study as reported:
1) The authors present a large number of observations of multiple parameters in a relatively small number of animals, in a very small number of herds. There is a lot of emphasis on the correlations between these parameters without providing any indication of why such correlations are of particular relevance to the diagnosis of infections of the mammary gland, nor how such analysis of the data provides advantages in the control of mastitis in herds. It would be far better if the study focussed on parameters that have direct relationships to mammary gland infection without being distracted by analysis of milk components that are largely dependent on other factors, such as nutrition and genetics.
Thank you for the recommendation. Although the most important milk parameter of our experiment was somatic cells count, our aim was to investigate other parameters as well. Through the analysis of other milk parameters, we wanted to highlight the fact that they should also be considered during the introduction of selective dry cow therapy. A compelling reason for inclusion was the metabolic disorders that arise because of inadequate feeding or stress and can thus significantly disrupt transition period. In our opinion, any attempts to reduce antibiotics should be based on continuous and comprehensive monitoring of appropriate indicators not only at farm but also at cow level. The implication is that not only the incidence of mammary gland infections should be low, but also low levels of disorders and diseases should be considered as another important aim.
2) The authors do not clearly separate the use of SCC in individual cows (or quarters) from SCC in bulk tank milk. While there two measurements are related, they are used in very different ways for mastitis control. Given that a major aim of the study is to explore the use of SCC as a basis for selective dry cow therapy, this is a significant shortcoming. The authors make several bold statements about the mastitis status in the small number of herds in the study, but provide no information that gives the reader insight into the details of mastitis in these herds. Despite comments such as “SCC obtained after calving on Farm 1, and Farm 3 are alarming…” I could find no information about causative bacteria or environmental challenge in these herds.
Thank you for pointing. In our research, the udder quarter was used as the unit of the analysis as mentioned in the Methods. We have not used the SCC of bulk tank test to assess the implementation of selective dry cow therapy.
We understand your concerns about the lack of detail regarding mastitis, however, we believe think that SCC and analysis of other milk parameters at dry-off and after calving are sufficient for this pilot study. We intend to identify the environmental and contagious mastitis pathogens, but only in the next step of our experiment, when we will test the methodology of selective dry cow therapy on all of three farms.
Some specific examples:
Table 6
Averaging of individual cow (or individual quarter) SCC is not wise, given the limits of quantification of SCC using the methods described. Precision of SCC counting is very poor at higher SCC. It would be better to describe the data in terms of proportion of cows which fall within categories of SCC.
Is 'n' the number of teats, cows, samples...? What information is available to explain the differences in numbers between the "Beginning of the dry period" and the "After calving" data?
Thank you for the recommendation. We reconsidered the inclusion of Table 6 in the manuscript and decided to remove it. We agree with you that the inclusion of these average of SCC of quarter milk samples was not wise. The table has been replaced by the following comment in L: 292 “More detailed statistics were compiled for SCC due to previous non-significant re-sults reported above. According to the arithmetic mean SCC values for each farm, it is clear that the dry-off and initiation of milk secretion management vary at the herd level. The extreme change in SCC values was recorded by Farm 1. At dry-off, the mean SCC value of quarter milk samples was 188,316 cells per mL. After calving, there was an increase of 448,434 to 636,750 cells per mL. If the SCC value of 200,000 cells per mL is considered as an indicator of bacterial infection [18], 30% of the teats at Farm 1 exceeded this thresh-old. After calving, this proportion decreased to 27%. The arithmetic mean SCC value at dry-off on the Farm 2 was 189,303 cells per mL with the proportion of teats above the 200,000 cells per mL threshold of 42%. At the beginning of lactation, Farm 2 experienced a large drop in the mean SCC and percentage of teats above the threshold. The arithmetic mean SCC was 94,966 cells per mL and the percentage of teats above the threshold for bacterial infection was 16%. The arithmetic means SCC values at dry-off and after calving on Farm 3 were 153,389 and 254,622 cells per mL, respectively. The proportion of teats above the SCC threshold of 200,000 cells per mL was 24% at dry-off on Farm 3. After calving, this proportion decrease to 18%.”
Line 247: Please provide some information about how these were done in the study herds. If you are extrapolating to the situation more broadly in the Czech national herd, please provide some indication of whether the study herds were representative. Are there any studies that describe how frequently various management approaches of relevance to mastitis, and specific mastitis control strategies, are implemented within Czech herds?
Thank you for pointing. The information about antibiotic application has been added in the method in L: 130 “The blanket dry cow therapy was the standard way to dry-off cow on all of farms”. As we mention in Material and Methods, herds were selected according to the typical breeding profile in the Czech Republic. We also considered the size of the herd to represent the average herd size in our conditions. In the Introduction we also mentioned that in 2019 more than 80% of dairy cows were dried-off by intramammary antibiotics, for this reason we included in the experiment dairy farms where only blanket dry cow therapy was applied. In addition, the farms were in the regions that are typical of the climatic and geographical conditions in the Czech Republic. We are not aware of any studies describing how frequently various management approaches of relevance to mastitis. Mastitis control strategies vary across the Czech farms and depend on the attending veterinarian.
Line 340: Blanket DCT is not a guarantee that bulk tank SCC will decline from one lactation to the next. These assertions need to recognise the multifaceted nature of mastitis infection within herds. This study seems to be drawing conclusions based on 3 herds with a paucity of information about mastitis control within these few herds.
Thank you for pointing. In our study, we did not work with bulk tank SCC, we analysed only quarter milk samples. The somatic cells count in L: 340 are the SCC averages of the quarter milk samples collected after calving, which were reported in Table 6. However, we realized that this way of phrasing the sentence was not appropriate, so we have changed it in L: 412.
Line 357: Ignores overlying recommendations based on herd BMCC
Recommendations based on herd BMCC were not included. In our research, the udder quarter was used as the unit of the analysis as mentioned in the Methods. We know that selective dry cow therapy can be performed according to SCC at herd level, cow level or quarter level, but to our knowledge the BMCC is more of a supporting indicator for selective dry-off at cow and quarter level. This is also mentioned in some studies that have established low BMCC for at least 12 months as a criterion for inclusion of herds in selective dry cow therapy at cow or quarter level. As mentioned above, we chose the quarter level as we believe this approach will be a more refined method for reducing the use of antibiotic in our conditions. However, for completeness of the farm descriptions, we decided to include in the Methodology the geometric means of the SCC of the individual herds for the period in which they were collected in L: 133 “The geometric mean of the bulk milk somatic cells count on each farm during the sampling period was: Farm 1 – 155,000 cells per ml; Farm 2 – 148,000 cells per ml; Farm 3 – 219,000 cells per ml.”
Line 371: How do you determine/define "prudently"?
Thank you for pointing. We agree that the used word was a bit confusing and unclear, and therefore it was removed. The changes can be found in L: 27 and 476.

Reviewer 4 Report
General comments:
The present publication present good objectives and data collection. However, from the results I missed the part of the title and objectives where you mentioned the reduction of antibiotic application. Per example: from the title I will expect that you will present in your data dairy cows that were exposed to antibiotics, one, two or even more times vs dairy cows that were not exposed to antibiotics that all, then see if all your variables measured in milk differed before and after calving. In such way your data could explain if the antibiotics has something to do with milk composition and quality, and more in the case of reporting antibiotics resistance cows… Are you able to classify your data with dairy cows that were treated or not with antibiotics? Then you should include this in your statistics and presents a P-value for this effect. Also, your title highlighted before and after “dry period”, but in fact it should be pre- and post-calving, I was wondering if due to high microbial contamination directly after calving in the dairy cows, this could increase susceptibility for mastitis (high milk SCC), but in your study does the type of mastitis (clinical or subclinical) was identified?
Specific comments:
L20-21: only prevention or treatment as well?
L89: Falcon tubes? Please provide the product description
L90-93: I am just curious how representative can be the sampling from the whole milk composition from the cow, maybe there is a gradient effect even adter remove the firsts 3 squirts.
L135: How many days before calving?
L 136: How many days after calving?
L140: I am just wondering why you did not present your dataset arranged by farm, instead or doing an average of the 3 farms for your variables during dry period and after calving. If you do so, this mean that all cows sampling were done more o less homogeneously across farms. I am also wondering if managements across farms in terms of length of days in milk, dry off, etc. were more or less homogeneous. I could assume that I long length milk dairy cows might develop more mastitis vs short days in milk period… maybe your data should show that…
Author Response
Dear Reviewer,
We are very pleased to have been given the opportunity to revise our manuscript. We also appreciate the time and effort you have dedicated to providing insightful feedback on ways to strengthen our paper. Our responses to your comments are mentioned line-by-line in red text in Word.
General comments:
The present publication present good objectives and data collection. However, from the results I missed the part of the title and objectives where you mentioned the reduction of antibiotic application. Per example: from the title I will expect that you will present in your data dairy cows that were exposed to antibiotics, one, two or even more times vs dairy cows that were not exposed to antibiotics that all, then see if all your variables measured in milk differed before and after calving. In such way your data could explain if the antibiotics has something to do with milk composition and quality, and more in the case of reporting antibiotics resistance cows… Are you able to classify your data with dairy cows that were treated or not with antibiotics? Then you should include this in your statistics and presents a P-value for this effect. Also, your title highlighted before and after “dry period”, but in fact it should be pre- and post-calving, I was wondering if due to high microbial contamination directly after calving in the dairy cows, this could increase susceptibility for mastitis (high milk SCC), but in your study does the type of mastitis (clinical or subclinical) was identified?
Thank you for your comment. In our experiment, we did not manage the application of antibiotics, but we mentioned several times in our manuscript that all farms used the blanket dry cow therapy. Following this, we evaluated the somatic cells count and other milk parameters to highlight the fact that selective therapy is not being implemented but could be. Thus, only cow with antibiotic treatment were included in the experiment and P-values describing the significance between cows that were treated with and without antibiotic cannot be published. Following your comments, we have made the title more specific: “Dynamics of Milk Parameters of Quarter Samples Before and After the Dry Period on the Czech Farms.”
And also the objective has been reformulated in L: 103 “As the quality of milk in the Czech Republic is increasing steadily in terms of hygiene and health indicators (composition, count of microorganisms, SCC, residues of inhibiting substances), we hypothesized that selective dry cow therapy could be applied to part of the teats. To investigate this, the aims of this study were to measure SCC and other milk parameters before and after dry period at the level of individual teats; assess the correlation between these values and explore the influence of parity, farms, time of evaluation, and day of calving were also examined to better understand possible variations in parameters.”
Quarter milk samples were taken at dry-off and after calving, not immediately before and after calving. We intend to identify type of mastitis and mastitis pathogens, but only in the next step of our experiment, when we will test the methodology of selective dry cow therapy on all of three farms. We comment on the increased susceptibility to mastitis after calving in the Discussion in L: 420.
Specific comments:
L20-21: only prevention or treatment as well?
Thank you for pointing. The sentence has been reformulated in L: 25 “However, this may change as our results indicate that the introduction of selective dry cow therapy is feasible, but well-known mastitis prevention and control strategies must be in place on farms.”
L89: Falcon tubes? Please provide the product description
Thank you for pointing. We added the product description in L: 136 “..into 50 mL falcone tubes (VWR International, Radnor, USA).”
L90-93: I am just curious how representative can be the sampling from the whole milk composition from the cow, maybe there is a gradient effect even adter remove the firsts 3 squirts.
The sampling procedure reflected the sampling principles specified in the Methodology of performance control for the Czech dairy Farms.
L135: How many days before calving?
The quarter samples were taken on the day of drying-off: 60 d before calving as mentioned in the Methods.
L 136: How many days after calving?
The quarter samples were taken between 6 and 21 d in milk as mentioned in the Methods.
L140: I am just wondering why you did not present your dataset arranged by farm, instead or doing an average of the 3 farms for your variables during dry period and after calving. If you do so, this mean that all cows sampling were done more o less homogeneously across farms. I am also wondering if managements across farms in terms of length of days in milk, dry off, etc. were more or less homogeneous. I could assume that I long length milk dairy cows might develop more mastitis vs short days in milk period… maybe your data should show that…
Thank you for pointing. Sampling was performed homogeneously from December 2021 to June 2022. The blanket dry cow therapy was the standard way to dry-off cow on all of farms. The length of the dry period was 60 d for all quarters of all cows. The number of days in milk was also balanced on all farms. Thus, length of lactation and dry period were not considered as effect.

Round 2
Reviewer 1 Report
I would like to thank the authors for their effort to address the previous comments. The manuscript is improved but there are still some concerns needs to be addressed:
Major concerns:
Ø The authors used multivariable GLM models to analyze their outcomes following the formula “Yijkl = μ + PARi + TIMEj + FARMk + b × (day of calving) + eijkl” and they used “the STEPWISE method, and the Akaike information criterion parameter were used to select appropriate effects for the model equation”. In addition the authors decided on their significance levels”. The significance levels were set at p <0.05 and p <0.01.”
· I wonder why the authors presented the results of the same models in 2 different tables (4 and 5) which is confusing to the reader.
· I would recommend that the authors to merge tables 4 and 5 into one table that follow a simple format as the one presented at Krattley-Roodenburg et al., 2021 (doi:10.3168/jds.2020-19133) representing the estimate values for each variable and the 95% CI.
· Are the models presented in tables 4 and 5 final models? If yes, Why the authors decided to keep non-significant variables in the final models (tables 4 and 5)? If not, I would recommend that the authors only present the final models.
· For table 4: I wonder why the authors represented the value of the F test for each variable?
· For tale 4: What is the authors explanation for the very small r square values for some models (SCC, titratable acidity, lactose, SNF)?
· For table 5: I would recommend only presenting the significant variables for each outcome.
· For all the outcomes (SCC, fat%, protein, etc.……):
o Did the authors test for normality assumptions for the outcomes? Because the there is a big variation in the values of the SCC among different farms, but it is not significant. That could be because the SCC is not normally distributed, or the data have numerous outliers and if the authors tried the log transformation that will satisfy the assumption and will clarify the effect.
Ø Conclusions: The current conclusion is beyond the scope of the study. The authors mentioned “It can be concluded that the collection and evaluation of milk quarter samples before and after dry period to obtain data important for the selective dry cow therapy is feasible on the Czech farms, however some farms are better suited for the initiation of selective dry cow therapy than other due to the mastitis prevalence.” How did the authors come to that conclusion from their study? The authors did not do any mastitis prevalence on the farms.
General comments:
Ø The authors tend to use the word teat in the manuscript to refer to the quarter. I would recommend that the authors change word “teat” into “quarter” in the manuscript as all the measurements were done at the quarter level and not the teat level.
Ø The language of the discussion needs to be edited for clarity. For example:
· I would recommend deleting “during lactation” line 280. As that contradicts the next statement “the dry period is defined as a non-lactation period.”
· It is not clear what the authors meant by “Many drying-off procedures are probably due to the differences in dairy farms. This is evident from our results”. What are the many drying off procedures from the current study? There is nothing in the study here about drying off procedures.
· line 287 “we compare” should be compared.
· line 293 “most volatile” what does that mean? In addition, reference 23 is not appropriate for the statement.
· Line 294 “which was confirmed by the regression included in statistical model” how was the seasonality confirmed in the study regression models?
· Line 295 “The significant differences between minimum and maximum” what does that mean? Did the authors test for significance between minimum and maximum values?
· line 296-297 “the GLM procedure confirmed that the content varied according to the farms included in the experiment” what did the authors mean by content?
· Line 298 “attributed to other factors” what are they?
· line 301” Metabolic disorders such as ruminal acidosis” ruminal acidosis is not a metabolic problem.
· Lines 307- 308 “slightly exceeds or equals the values of the selected studies”. Which selected studies? The authors did not mention that they selected any studies.
· Line 416-417 “collection and evaluation of milk quarter samples before and after dry period to obtain data important”
Ø Line 144: Did the author consider 0.8 or -0.8 as moderately strong or very strong? Please edit the statement for clarity.
Ø Line 170: I would recommend deleting “As be seen from the table,” and start the statement with Arithmetic mean.
Ø Line 172: I would recommend deleting “our milk samples taken at different times,”
Ø Line 174: “Losses of milk fat and protein after calving” The term losses in the statement is not accurate as there were no losses occurred. The authors can use the term difference as they are comparing two different stages without accounting for other parameters that will influence the outcome such as DIM and the volume of milk produced as it is well known that there are differences in the milk components along the lactation cycle (fresh vs peak vs late).
Ø Line 178. “physical properties”. This is the first mention for this term please add the measured physical properties between parentheses.
Ø For the table 1: it is repetition of no value to present both the SD and SE for the measured parameters. Please keep only one of them and I would recommend keeping the SE.
Ø It is not clear how the authors handled the missing values when they did Pearson’s correlation coefficient? For example:
· At dry off for fat% only 193 samples were measured while for SCC 211 samples.
· After calving 154 samples (fat%) while SCC was 175 samples.
Ø I still recommend adding tables 2 and 3 as supplement.
Author Response
Thank you for all your comments.

Reviewer 3 Report
Thank you for your responses to my previous comments.
Author Response
Thank you for your comments and final decision.
Reviewer 4 Report
Thanks for considering the suggestions
Author Response

(The authors gave the same response as above.)

Round 3
Reviewer 1 Report
I would like to thank the authors for addressing most of my comments. I will defer to the editor for the following:
- Keeping non-significant variables in the final models.
- Combining tables 4 and 5.
- Adding tables 2 and 3 as supplement.
Author Response
Dear Reviewer and Editors,
Thank you for allowing us to revise our manuscript again according to your comments. First, the Simple Summary has been reformulated based on your requirement. The summary is now more unified with the aim and the Conclusion. The keyword “Antibiotic treatment” has been deleted and replaced with “Selective dry cow therapy.” Three comments have been reconsidered, and responses to them are mentioned below in red text. We have made a minor spell check throughout the manuscript, and edited one of the sentences in the Introduction because we thought it was misleading.
Best regards,
Veronika Legarová
